# An Online Charging Scheme for Wireless Rechargeable Sensor Networks Based on a Radical Basis Function

**DOI:** 10.3390/s20010205

**Published:** 2019-12-30

**Authors:** Jia Yang, Jian-Shuang Bai, Qiang Xu

**Affiliations:** 1Chongqing Energy Internet Engineering Technology Research Center, Chongqing University of Technology, No. 69 Hongguang Avenue, Chongqing 400054, China; 2College of Computer Science and Technology, Chongqing Technology and Business University, No. 19 Xuefu Avenue, Chongqing 400067, China

**Keywords:** wireless rechargeable sensor network, online charging schemes, RBF neural network, the dynamic energy consumption rate, energy hole rate

## Abstract

The node energy consumption rate is not dynamically estimated in the online charging schemes of most wireless rechargeable sensor networks, and the charging response of the charging-needed node is fairly poor, which results in nodes easily generating energy holes. Aiming at this problem, an energy hole avoidance online charging scheme (EHAOCS) based on a radical basis function (RBF) neural network, named RBF-EHAOCS, is proposed. The scheme uses the RBF neural network to predict the dynamic energy consumption rate during the charging process, estimates the optimal threshold value of the node charging request on this basis, and then determines the next charging node per the selected conditions: the minimum energy hole rate and the shortest charging latency time. The simulation results show that the proposed method has a lower node energy hole rate and smaller charging node charging latency than two other existing online charging schemes.

## 1. Introduction

Wireless sensor networks (WSNs) are being widely used in environmental monitoring [1], forest fire warning [2], and medical care [3], with the vigorous development of wireless communication technology, sensor technology, and microelectronic technology. However, the energy supply modes for sensor node are restricted, which seriously affects the performance and development of sensor networks. Scholars have conducted considerable research for the sensor network energy problem. These works can be roughly divided into three categories: energy saving methods [4,5], energy collection methods [6], and wireless charging methods [7]. Energy saving methods usually sacrifice a certain amount of network performance and the energy of sensor nodes limits the increase in network lifetime. The energy conversion method has low energy conversion efficiency and uncontrollable energy acquisition, and accurate prediction is difficult. The wireless charging method is equipped with an active charging power node in the network, and the static or mobile charging node actively provides the sensor node with efficient and timely charging services. The charging process is controllable and predictable. Recent advances in wireless charging techniques [8] and rechargeable lithium-ion battery technology [9] have produced a new solution for the energy problem of wireless sensor networks, and wireless rechargeable sensor networks (WRSNs) [10,11,12] have emerged.

The planning and research of mobile charging equipment can effectively improve the charging efficiency and help to find the best node energy supplement scheme due to the high cost and limited energy of wireless charging equipment [13]. Most of the discussion and research on WRSN charging schemes are focused on offline charging. Offline charging assumes that the energy consumption rate of each sensor node is always fixed. The mobile charger (MC) plans the charging path in advance according to the energy consumption of the previous operating phase of the nodes in the network. During the charging process, the MC periodically follows the planned path option to replenish energy to the sensor nodes. However, the energy consumption of the sensor nodes in the actual situation is highly dynamic and diverse because WSNs are usually deployed in the monitoring area in close contact with the surrounding environment [14], so the energy consumption rate of the nodes is not fixed. Invariably, MC charging according to the pre-defined charging path and scheme in offline charging will produce a serious energy hole in the sensor node and a serious decrease in the performance of the charging scheme, which is not suitable for actual sensor network environments.

Some researchers proposed online charging schemes to address the shortcomings of offline charging. In online charging, the MC dynamically formulates a charging scheme in real time according to the actual remaining energy status of the sensor node and it quickly responds to the node’s charging request. Online charging is more suited to the actual node’s energy requirements and changes when compared with offline charging. However, the existing related online charging work does not fully consider the timely charging of nodes and the reasonableness of MC’s response to node charging requests, which also causes node energy holes in the charging process [15]. The problem of node energy holes is particularly obvious when many targets must be monitored in the network and the number of nodes requesting charging increases.

These studies rarely considered estimating the dynamic energy consumption rate to determine the optimal charging request threshold and the next charging node. Therefore, an online charging scheme based on a radical basis function (RBF) for the energy hole avoidance of WRSN is proposed here to reduce the energy hole rate [16] and the waiting time of charging-needed nodes. The RBF is used to predict the dynamic energy consumption rate to improve the real-time ability and fairness of the node charging response. The next charging node is selected based on the minimum energy hole rate and the shortest charging completion time, which considerably reduces the energy hole rate of nodes and the waiting time of the node to be charged. The findings provide four main contributions:(1)An RBF prediction model for dynamic energy consumption rate is proposed. The energy consumption of each node is different and it changes at any time due to the differences in node location, environment, and data transmission intensity. The dynamic energy consumption rate of each node is estimated based on the RBF neural network, which can ensure the real-time acquisition of node energy consumption.(2)A dynamic evaluation model was designed for the optimal charging request threshold. The charging request can only be sent to the MC when the residual energy of the node is lower than the threshold value of the charging request. We used the energy consumption data predicted by RBF to theoretically analyze and estimate the threshold value of charging request to reduce the waiting time of the charging-needed nodes and considerably improve the fairness of node charging response starting from the three constraints of node energy consumption, network residual energy limit, and MC average service time.(3)The next charging node is selected in real-time. The node that minimizes the number of energy holes in the network and takes the shortest time to complete charging is always selected as the next charging node by comparing the charging waiting time of the charging-needed nodes, avoiding energy holes in the nodes as much as possible and reducing the waiting time of the nodes needing charging.(4)We verified the accuracy and reliability of the online charging scheme and explored some influencing factors through theoretical analysis and simulation research. The experiments showed that the radical basis function-energy hole avoidance online charging scheme (RBF-EHAOCS) proposed in this paper fully proves its performance and optimization effect in terms of network energy hole rate and the charging latency of the charging-needed nodes.

The rest of this article is organized, as follows: Section 2 reviews related work. Section 3 introduces the problem description and system model. In Section 4, we propose the online charging scheme for energy starvation avoidance in detail. The analysis and simulation results of influencing factors are outlined in Section 5. Section 6 provides the conclusion of this article.

## 2. Related Work

The WRSN online charging scheme refers to the scheme of MCs to determine the next charging node in real time according to the residual energy and dynamic energy consumption rate of the node, and to plan charging based on demand. He et al. [14] proposed a nearest-job-next with preemption scheme (NJNP), which always chose the nearest rechargeable node to MC as the next recharging node, which reduced the charging cost of MC in the charging process, but ignored the fairness of the charging response, which easily led to a long distance from MC. Nodes that are far from the MC must wait long times for on-demand charging service, which can easily produce the energy holes. Wang et al. [17] converted the charging scheduling problem into an optimized profit traveling salesman problem with energy constraints and charging deadline limits. However, the energy consumption and maintenance costs of the scheme markedly increased due to the use of network clustering and a separate data collection vehicle. The assumption that the energy consumption rate of sensor nodes with the same number of hops from the base station is not in line with the situation in reality. Feng et al. [18] showed that the online charging process should try to avoid the node plunging into energy holes, and they proposed an energy replenishment scheme for hunger avoidance. However, this scheme ignores the influence of the MC carrying energy on the actual charging route recommendation, directly assumes that the MC energy is infinite, and does not consider the influence of the change of the node energy consumption rate on the residual energy value of the actual node, which results in the occurrence of energy holes due to the unreasonable charging route recommendation. Lin et al. [19] proposed a charging scheduling scheme, called time and distance priority (TADP), which quantitatively calculates the priority of charging tasks for charging planning. However, it assumes that the energy consumption of each node is the same, which does not conform to the diversity of energy consumption changes in the nodes in actual situations. Zhu et al. [20] proposed the energy starvation avoidance online charging scheme, which aims to minimize the network starvation node ratio while also reducing the MC charging cost and node average charging delay. However, the scheme performs an iterative calculation on the request transmission threshold based on the energy consumption rate at the initial time, and it does not fully consider the dynamic change in the node energy consumption rate at the next moment, which can easily occur when the threshold in the network is too large or too small, which causes the network nodes to be prone to energy holes. Other nodes do not record or transmit their own remaining energy during the charging of the MC, which means that the energy holes for other nodes in the charging process cannot be detected in time. Figure 1 summarizes these related works.

We found that most of the existing online charging schemes determine the charging order of the nodes needing charging without considering the different energy consumptions and the instant changes in each node, which reduces the overall performance of the network and causes nodes to fail to receive charging service in time. Some authors assumed that the energy capacity that is carried by the MC is infinite and the energy loss in the movement of the MC was not considered, which are not in line with the actual situation. In an actual network, the threshold value of a node’s request for charging should also be corrected in real-time as the remaining energy of the network node changes to minimize the charging waiting time of the node. The fairness of the charging response of the scheme can be optimized when the three factors of the adaptive threshold of charging request, the constraints of MC energy, and the dynamic energy consumption rate are all satisfied. In this paper, we propose an energy hole avoidance online on-demand charging scheme that is based on RBF, named RBF-EHAOCS, to allow for the MC to quickly and effectively replenish the charging-needed nodes in the network. Constraints and charging schemes, combined with the actual dynamic changes of the nodes in the network, were designed to ensure the long-term reliable operation of a WRSN.

## 3. Problem Description and Network Model

### 3.1. Problem Description

In the actual WRSN, the residual energy of each node changes in real time, and the change trends are different, due to different data transmission tasks. Estimating the dynamic energy consumption rate of nodes, determining the charging request threshold value, and ensuring the charging requirements of the nodes in the charging service waiting area are quickly met and they are charged before the node’s energy is exhausted to avoid node energy holes as much as possible, are the keys for completing the charging planning. In this paper, we propose an online charging scheme for avoiding energy starvation (RBF-EHAOCS) to plan the charging process, with the main purpose of solving the following three problems:(1)Different nodes transmit different amounts of data at different times, which result in different energy consumption conditions for each node. We wanted to efficiently and quickly predict the dynamic energy consumption rate of the nodes to meet the dynamic changes in the energy consumption needs in actual situations.(2)Our method updates the charging threshold immediately before selecting the next charging node to prevent other charging-needed nodes with energy holes when MC charges a certain node. The problem was determining how to estimate the threshold value of the node’s request for charging.(3)The MC can only perform one-to-one charging and carries limited energy. Therefore, we considered how the MC determines the next charging node under the condition of limited charging service time, and ensures that it can smoothly return to the service station (SS) to supplement energy at low energy to improve the fairness of the charging response.

The energy hole rate of the network can be reduced, the network life can be lengthened, the waiting time of the charging-needed nodes before receiving the charging services can be reduced, and the best applicable conditions of the scheme can be framed by solving the above problems.

### 3.2. Network Model

The proposed network model of the online charging system adopts a square structure, that is, the *N* rechargeable sensor nodes of the network are randomly distributed in a square area with side length *D*, and a base station (BS), and a SS are both defined as located in the center of the square area.

The network model is mainly represented as $\left(A,W,E,P,R_{A},D\right)$. A={A(1,1),…,A(i,j)} is not only the set of locations of *N* sensor nodes in the network, but it is also the definition of each node. $W=\left\{d_{ij}|\left(i,j\right)\in A\left(i,j\right)\right\}$ is a set of weights and $d_{ij}=Eudis\tan ce\left(A\left(i,j\right),A\left(i^{\prime },j^{\prime }\right)\right)$ represents the Euclidean distance between node A(i,j) and node $A\left(i^{\prime },j^{\prime }\right)$. Through the collection of *A* and *W*, the specific distribution of the entire network can be known [21]. *E* is the loadable energy of each sensor node and *P* is the energy capacity that is carried by the MC. $R_{A}=\left\{R_{A\left(i,j\right)}|A\left(i,j\right)\in A\right\}$ is the set of dynamic energy consumption rates of each sensor node. The dynamic energy consumption rate is predicted by the RBF neural network to quickly meet its dynamic dynamics. A single MC with a fixed charging efficiency of η is initially located in the SS. After receiving the charging request, it moves at a constant speed *v* in the network region. The energy loss per unit distance of the motion is *c*. The sensor nodes have a fixed transmission radius *R*, and they periodically sense the remaining power of the node and generate data transmission to the base station, and simultaneously issue a charging request and receive feedback from the base station when the remaining power is below the threshold. Figure 2 shows the network model of the entire system.

The MC of the system mainly has four states in conjunction with the sensor network to realize online charging: idling, moving, charging, and regression. The specific working process of the system is as follows: each sensor continuously monitors the information of the area and its own remaining power value, and sends a sensing message to the BS every time interval Δt while using multi-hop communication. When the remaining power value is lower than the charging request threshold, it will send a charging request to the BS. The BS sends the received charging request directly to the MC after receiving the energy notification and charging request of the node, which is initially in an idle state, via long-distance communication. The MC establishes a charging request message for maintaining a charging service waiting area to store the charging-needed node whose remaining power is lower than the threshold, and then selects the next charging node through the RBF-EHAOCS and moves to the node, and then enters the charging state. When the MC completes charging and the charging service waiting area is cleared or its own energy is insufficient, the MC enters a regression state and then returns to the SS for supplemental energy. Figure 3 shows the state transition relationship of the MC.

## 4. Design and Implementation of the RBF-EHAOCS

### 4.1. Estimating Dynamic Energy Consumption Rate of Each Node by RBF Neural Network

Weighted average method [22], RBF neural network [23,24], and the evolutionary neural network [25,26] are the available methods for the prediction of dynamic energy consumption rate. The weighted average method is more focused on long-term trend changes, which is not conducive to the timeliness of calculation [27]. In an actual WSN, the energy consumption of some nodes acting as the main cluster head changes more, which might lead to large deviation between the predicted value and the actual value while using this method. The evolutionary neural network is an organic fusion of evolutionary computing and neural networks, which is more suitable for intelligent learning and prediction in the context of large systems and big data [28]. Complex and time-consuming problems with adaptive processes are prone to occurring in the WSN with small data volumes and small network scales, which might prevent the real-time performance of the algorithm from being guaranteed. The RBF neural network is used as a nonlinear multi-layer single feedforward local approximation network [23] that has strong robustness, nonlinear fitting ability, and memory ability. RBF has simpler learning rules, shorter prediction time, and better fitting effect on energy consumption when compared with other two predictive learning methods [29]. Therefore, we chose the RBF neural network for predicting the dynamic energy consumption rate of the nodes.

Its radial basis functions are commonly classified into Gauss function, anomalous s-type function, and quasi-multiple quadratic function [30]. The RBF neural network that is selected in this study is defined as a generalized network. The input quantity is the remaining power $RE_{A}$ of each sensor node at the current time *t* and the output is the dynamic energy consumption rate $R_{A\left(i,j\right)}$ of each node. Figure 4 shows the specific topology of the RBF neural network.

The basis function of the RBF neural network in this paper uses the Gauss function, that is, the hidden layer activation function $\phi \left(RE_{A}-X_{m}\right)$ is:(1)$$\phi \left(RE_{A}-X_{m}\right)=\exp\left(-\frac{1}{2\sigma }RE_{A}-X_{m}\right)$$
where $X_{m}$ represents the central vector of the hidden layer node, $RE_{A}-X_{m}$ is the European norm of the remaining power and the hidden layer center, and the standard deviation of learning is:(2)$$\sigma =\frac{d_{max}}{\sqrt{2}m}$$
where the number of cluster centers (the number of hidden layer nodes) is *m* and $d_{max}$ represents the maximum Euclidean distance between cluster centers. Subsequently, the unsupervised K-means clustering scheme [31,32] is used to initialize the parameters and realize the self-organizing selection center. The least mean square (LMS) scheme calculates the learning weights [33] and finally, the standard deviation normalization method performs the output anti normalization. The conversion function is:(3)$$R_{A}^{\prime }=\left(R_{A}-\mu \right)/\sigma$$
where μ is the mean of all sample data, $R_{A}$ is the dynamic energy consumption rate predicted by the RBF neural network, and $R_{A}^{\prime }$ is the energy consumption rate of the output after normalization.

As shown in Figure 5, the RBF neural network can accurately predict the dynamic energy consumption rate of each sensor node at a certain moment, which meets the diversity and dynamic requirements of the energy consumption rate in actual situations, and it helps to reduce the energy hole rate of nodes in the network.

### 4.2. Estimating Threshold Value Range of Charging Request $E_{thred}$

When the sensor node perceives that its remaining power is lower than the charging request threshold $E_{thred}$, it sends a charging request to the BS while also transmitting the charging request [34]. The value of the charge request threshold $E_{thred}$ significantly impacts the performance of the charging scheme. If the selected charging request threshold $E_{thred}$ is too high, the remaining power of the node is still very large when the MC moves to the node to be charged, and the frequency that is generated by the charging-needed node increases, which causes the MC to lose too much energy during the movement, which reduces the energy use of the MC and does not meet the requirements for reducing energy consumption. Conversely, if the selection is too low, the maximum charging waiting time after the node sends a charging request to the BS is limited, which leads to energy holes in the node in advance when the MC has not moved to the node in time. The size of the charge request threshold $E_{thred}$ is estimated from the following three aspects:
(1)For the energy consumption of a single node, the remaining energy of the node when transmitting the request must ensure that energy holes do not appear in the node before the MC moves to the node with the shortest delay and provides charging service. While assuming that the MC is initially moved at the location of the SS, the shortest delay $t_{ij}$ for the MC to move to node A(i,j) is:(4)$$t_{ij}=t\left(MC,A\left(i,j\right)\right)=\frac{Eud\left(MC,A\left(i,j\right)\right)}{v}$$
where Eud(MC,A(i,j)) is the Euclidean distance of the MC from SS to the node A(i,j) and v is the moving speed of the MC. The node location information can be provided by the location set *A* of the network model, which is available from the first estimation principle:(5)$$t_{ij}<\frac{E_{thred}}{R_{A\left(i,j\right)}}$$
where Eud(MC,A(i,j)) represents the dynamic energy consumption rate of the next moment that the node A(i,j) currently predicts through the RBF neural network, and it can be taken from the $R_{A}$ parameter of the network model. All of the sensor nodes in the network must satisfy Equation (5), so:(6)$$E_{thred}>max\left[\frac{Eud\left(MC,A\left(i,j\right)\right)}{v}\times R_{A\left(i,j\right)}\right]$$

Let the position set *A* of all nodes be a matrix of a×b, and then i and j in the above formula should satisfy 1≤i≤a and 1≤j≤b.

(2)For the overall energy status of the nodes in the network, the MC determines the time to complete charging of the charging-needed node before the moment and the time required for the MC to move from the current node to the node, which constitutes the MC charging of the node. The remaining power of the node should ensure that the node remains alive during the MC charging service waiting time. Subsequently, the MC charging service waiting time $t_{w}$ of each node is:(7)$$t_{w}=\{\frac{d}{v}+\left(\frac{2d}{v}+\frac{E-E_{thred}}{\eta }\right)+\cdots +\left[\frac{Nd}{v}+\frac{\left(N-1\right)\left(E-E_{thred}\right)}{\eta }\right]\}/N$$
where $E-E_{thred}$ represents the energy to be replenished by the charging-needed node, η is the charging efficiency of the MC, and d represents the average value of the Euclidean distance between the sensor nodes, which can be calculated by the weight *W* parameter $d_{\Delta ij}$ of the network model:(8)$$d=\frac{1}{N^{2}}\sum_{i=1}^{a}\sum_{j=1}^{b}d_{\Delta ij}$$

While substituting Equation (8) into Equation (7) and simplifying it, we obtain:(9)$$t_{w}=\frac{\left(N+1\right)\sum_{i=1}^{a}\sum_{j=1}^{b}d_{\Delta ij}}{2vN^{2}}+\frac{\left(E-E_{thred}\right)\left(N-1\right)}{2\eta }$$

During the MC charging service waiting period, the remaining power of the node should be greater than the energy consumption value based on the average node energy consumption rate to ensure that the energy hole rate of nodes is minimized, that is:(10)$$E_{thred}>t_{w}\times R$$
where the average node energy consumption rate R should be expressed as:(11)$$R=\frac{\sum_{i=1}^{a}\sum_{j=1}^{b}R_{A\left(i,j\right)}}{N}$$

Substituting Equations (9) and (11) into Equation (10) and then simplifying it, we obtain:(12)$$E_{thred}>\frac{\sum_{i=1}^{a}\sum_{j=1}^{b}R_{A\left(i,j\right)}}{N}\times \left[\frac{\frac{\eta \left(N+1\right)\sum_{i=1}^{a}\sum_{j=1}^{b}d_{\Delta ij}}{vN^{2}}+E\left(N-1\right)}{2\eta +\left(N-1\right)\sum_{i=1}^{a}\sum_{j=1}^{b}R_{A\left(i,j\right)}/N}\right]$$

(3)For the energy capacity carried by the MC, the number of nodes that perform the primary charging service when the MC is fully loaded with energy is not too small; otherwise, the energy use of the MC is reduced and unnecessary energy loss increases. The average number $N_{ps}$ of nodes for a single charge service with MC full load energy is:(13)$$N_{ps}=\left[\frac{P-dc}{dc+E}\right]$$
where c denotes the energy of the MC moving unit distance required loss. Let α(0<α<1) be the desired MC energy use rate, and the average arrival rate λ of the charging request is expressed as:(14)$$\lambda =\frac{\sum_{i=1}^{a}\sum_{j=1}^{b}R_{A\left(i,j\right)}}{E-E_{thred}}$$

Afterwards, under the condition of the average arrival rate λ, the number of nodes that can be served should be greater than the expected number of MC average service nodes, that is:(15)$$N_{ps}\alpha \leq \lambda TN$$
where T represents the average service time for the MC to charge the node when the average arrival rate λ is expressed as:(16)$$T=\frac{\left(\frac{d}{c}+\frac{E}{\eta }\right)+\left(\frac{2d}{c}+\frac{2E}{\eta }\right)+\cdots +\left(\frac{Nd}{c}+\frac{NE}{\eta }\right)}{N}=\frac{\left(N+1\right)}{2}\left(\frac{d}{c}+\frac{E}{\eta }\right)$$

While substituting Equations (13), (14), and (16) into Equation (15), we obtain:(17)$$E_{thred}\geq \frac{\alpha \left(N+1\right)\left(\frac{d}{c}+\frac{E}{\eta }\right)\sum_{i=1}^{a}\sum_{j=1}^{b}R_{A\left(i,j\right)}}{2\left[\frac{P-dc}{dc+E}\right]}$$

According to the above three aspects, the lower limit of the charging request threshold should be the maximum value in Equations (6), (12), and (17). Other nodes update the remaining energy situation, while the MC is providing the charging service for a charging-needed node, so that a new round of real-time estimation of the charging threshold will be conducted before each selection of the next charging contact. At this time, the range of values of $E_{thred}$ can significantly improve the energy use rate of the MC while reducing the energy hole rate of the nodes. Algorithm 1 shows the pseudo code of the scheme.
**Algorithm 1**. Algorithm for selecting the charging threshold $E_{thred}$1. While (RBF working status = 1) **do**2. {Initialization: BS&SS position (P.x1, P.y1) = (sink.x, sink.y); N = 100; Nc = 200; P = 5000; E = 10; c = 0.004; U = 0.8;3. Function EUCLD: EudMC = EUCLD(P.x,P.y,Pt.x,Pt.y);4. Calculate EudMC according to function EUCLD;5. **for** a1 = 1:size(EudMC,1)6. **for** b1 = 1:size(EudMC,2)7. Calculate T(a1,b1) according to Equation (4);8. Calculate Ethred_1(a1,b1) according to Equation (5);9. **end for**10. **end for**11. [max1,~] = max(max(Ethred_1)); // Obtain the first value12. Calculate Ddist according to Equation (8);13. RA = load(Energy consumption rate obtained by RBF);14. Calculate the average of node energy consumption rate: Rdist = (1/N) × (sum(RA(:)));15. Calculate Ethred_2 according to formula (12);16. [max2,~] = max(max(Ethred_2)); // Obtain the second value17. Calculate Nps according to Equation (13);18. Calculate Tps according to Equation (14);19. Calculate Ethred_3 according to Equation (17);19. [max3,~] = max(max(Ethred_3)); // Obtain the third value20. Ethred = max(max(max1,max2),max3); // Take the max value21. }

### 4.3. Next Charging Node Selection Scheme

MC selects the next charging node during the movement process, which is key in guaranteeing charging performance. For selecting the next charging node with RBF-EHAOCS, if the charging service waiting area is not empty, for all of the charging-needed nodes, the current maximum charging waiting time of the node is calculated while using the dynamic energy consumption rate predicted by the RBF. When the other node is selected as the next charging node, the minimum charging waiting time of the node compares the answers that were obtained by the two groups and always selects the node with the least number of node energy holes in the network as the next charging node [20]. When the next charging node is specifically selected, the node that can be charged the fastest is selected as the next charging node in the charging candidate node set to minimize the charging latency of other requesting charging nodes. Figure 6 shows the scheme flow chart. The specific process is mainly divided into the following five steps:
(1)Before the MC starts charging scheduling, the current maximum charging waiting time of each node to be charged in the charging service waiting area is first calculated [35]. For example, the maximum charging waiting time of the node A(i,j) to be charged is expressed as:(18)DelayA(i,j)(t)=REA(i,j)RA(i,j)+tsA(i,j)−t
where $RE_{A\left(i,j\right)}$ represents the remaining power of node A(i,j) at this time, $R_{A\left(i,j\right)}$ is the dynamic energy consumption rate of the node at the current time, $ts_{A\left(i,j\right)}$ is the time when node A(i,j) sends a charging request, and t is the current time. No energy hole exists in node A(i,j) if $Delay_{A\left(i,j\right)}\left(t\right)>0$; otherwise, an energy hole exists and the node needs to be removed from the set.(2)For the charging-needed node that does not have an energy hole in the charging service waiting area, the MC sequentially calculates the minimum charging waiting time of the node when another node is selected as the next charging node. For example, when node A(i,j) is the next charging node, the minimum charging waiting time of node $A\left(i^{\prime },j^{\prime }\right)$ is expressed as:(19)$$W\left(A\left(i,j\right),A\left(i^{\prime },j^{\prime }\right)\right)=t_{\left(MC,i\right)}+\frac{E-E_{A\left(i,j\right)}-R_{A\left(i,j\right)}\times \left(t-ts_{ij}+t_{\left(MC,i\right)}\right)}{\eta }+t_{\Delta ij}$$
where $t_{\Delta ij}$ represents the time required from node A(i,j) to node $A\left(i^{\prime },j^{\prime }\right)$. The specific expression is:(20)$$t_{\Delta ij}=t\left(A\left(i,j\right),A\left(i^{\prime },j^{\prime }\right)\right)=\frac{d_{\Delta ij}}{v}$$

If $Delay_{A\left(i,j\right)}\left(t\right)\geq W\left(A\left(i,j\right),A\left(i^{\prime },j^{\prime }\right)\right)$ is established, which indicates that node A(i,j) is the next charging node, node $A\left(i^{\prime },j^{\prime }\right)$ does not have an energy hole. If node A(i,j) satisfies the inequality for any remaining charging-needed nodes, indicating that the node A(i,j) is the next charging node, the charging service waits. No energy holes appear in other charging-needed nodes in the zone. At this time, node A(i,j) is added to the charging node candidate set Z, and all of the nodes meeting the above conditions are found. If the set Z is empty, a charging pending set *W* is established for each node to be charged in the charging service waiting area. When a node is identified as the next charging node, the number of nodes and the identification number in the case of $Delay_{A\left(i,j\right)}\left(t\right)<W\left(A\left(i,j\right),A\left(i^{\prime },j^{\prime }\right)\right)$ are stored in *W.*

(3)The next charging node is selected: (1) If the set Z is not empty, the MC calculates the time that is required to complete the charging when each node is selected as the next charging node $T_{ch}$, which is:(21)$$T_{ch}=t_{\left(MC,i\right)}+\frac{E-E_{A\left(i,j\right)}\left(t+t_{\left(MC,i\right)}\right)}{\eta }$$

The next charging node needs to ensure that the remaining power of the MC is sufficient for the MC to return to the base station to replenish energy after completing the charging task. The following inequalities need to be verified:(22)$$RE_{\left(MC,t\right)}-c\times Eud\left(MC,A\left(i,j\right)\right)-E+E_{A\left(i,j\right)}\left(t+t_{ij}\right)\geq c\times Eud\left(A\left(i,j\right),SS\right)$$
where $RE_{\left(MC,t\right)}$ represents the residual energy value of the current moment of the MC and Eud(A(i,j),SS) is the Euclidean distance of the node A(i,j) to the SS. If Equation (22) is satisfied, the node with the shortest charging time is selected as the next charging node. (2) If the set *Z* is empty and a node is the next charging node, the number of nodes satisfying $Delay_{A\left(i,j\right)}\left(t\right)<W\left(A\left(i,j\right),A\left(i^{\prime },j^{\prime }\right)\right)$ is the smallest, and the node can be used as the next charging node to minimize the number of nodes with an outgoing energy hole. (3) If the next charging node is selected according to condition (1) or (2) in this paragraph, Equation (22) is not satisfied, and the charging-needed node that is closest to MC is selected as the next charging node to determine whether Equation (22) is satisfied. The nearest node to be charged by the MC is the next charging node. (4) If the next charging node that meets the requirements is not found according to conditions (1)–(3); in this paragraph, the remaining energy of the MC is insufficient, and the base station compensation energy is immediately returned. If the selection of the next charging node has been completed in step (3), normal charging is performed, the node is deleted in the charging service waiting area, and the collection Z is cleared after the charging is completed. If step (3) still fails to select any candidate node, the MC returns to the SS to replenish the energy and then proceeds to step (5). (5) Repeat the first four steps until the charging service waiting area is empty, and the MC then returns to the SS and replenishes its own power.

## 5. Simulation Experiment and Performance Analysis

### 5.1. Simulation Environment and Parameter Settings

In this section, the simulations and tests, based on the MATLAB 2016b simulation platform (MathWorks, Natick, MA, USA), of the online charging scheme that is proposed in this paper are described. We compared the performance of RBF-EHAOCS with the typical online charging scheme NJNP [14] and Starvation Avoidance Mobile Energy Replenishment (SAMER) [18].

In the simulation experiment, the simulation area was a square area with a side length of 200 m. The random network was used for spreading and the number of nodes to be simulated was 100. The original data set that was used by the RBF neural network was divided into two parts: the training set and the test set. The training set contained 700 samples and the test set contained 300 samples. The data generation of each sensor node followed a Poisson process with an average arrival time interval of 50 s, and the network bandwidth was 10 Kbps [36]. The entire simulation duration was 72,000 s. Table 1 provides the other test parameters and corresponding default values.

We analyzed the performance of the WRSN online charging scheme while using the following two technical indicators:(1)Energy hole rate of the nodes is defined as the ratio of the number of nodes to be charged with the energy hole and the total number of nodes to be charged. It is an important index that is used to evaluate the performance of the charging scheme. The smaller the energy hole rate of the nodes, the better the performance of the charging scheme.(2)Charging latency of the charging-needed node is defined as the time interval between when the charging-needed node sends a charging request and when the MC starts charging this node. The smaller the charging latency of the charging-needed node, the more fair the charging response, and the higher the reliability of the charging scheme [35].

### 5.2. Network Performance under Different Schemes

We verified the reliability and optimization of our scheme by quantifying the two technical indicators mentioned above under the preset experimental simulation parameters. Table 2 shows the specific experimental data.

RBF-EHAOCS has a lower energy hole rate and charging latency than NJNP and SAMER. The charging-needed node under NJNP needs to wait a long time for charging service after receiving the charging request due to the lack of real-time estimation of the charging threshold and the overly simple selection strategy of the next charging node. During the charging waiting period, energy holes are more likely to occur due to the unreasonable planning of the charging sequence. Relatively, SAMER uses the remaining energy of nodes in the previous moment for charging planning. SAMER cannot adapt to the actual situation in in real-time and the performance is not good enough as a result when compared with RBF-EHAOCS, which directly uses RBF to predict the energy consumption.

In an actual network, three factors (number of nodes, amount of energy capacity carried by MC, and charging efficiency between MC and the charging-needed node) all affect the experimental results. Hence, we compared the effects of each scheme under these three influencing factors, and explain the superiority of RBF-EHAOCS in detail.

### 5.3. Network Performance under Different Node Numbers

The other parameters were maintained at the default value and the number of network nodes was gradually increased from 25 to 200 to explore the impact of the number of network nodes on the performance of the scheme. Figure 7a–c show the network performance changes.

Figure 7a shows that, when the number of nodes is 75 or less, since the number of nodes is small at this time, the proportion of nodes needing charging in the network is also low, and the MC can quickly complete charging for a small number of nodes to avoid energy holes. The number of nodes with energy holes in the three schemes increases as the number of nodes increases. The higher the number of nodes, the higher the number of charging-needed nodes that must be provided with MC charging services, the higher the charging load of the MC. The charging service will become more frequent when the node fails to receive the charging service in time, and the energy hole rate of the nodes will also increase. However, during this process, RBF-EHAOCS uses RBF to effectively predict the node consumption rate while using the dynamic characteristics of the node consumption. Based on this, the request charging threshold of the charging-needed node is selected and minimized. The energy hole rate of the node is the next charging node, so the energy hole rate of nodes under this scheme is lower than with NJNP and SAMER. When the number of nodes is around 165, the energy hole rate of nodes with SAMER is higher than with NJNP. SAMER cannot quickly and efficiently select the next charging node because the number of nodes is too large, which results in a rapid increase in nodes with energy holes.

As shown in Figure 7b, the charging latency of the charging-needed node of the three charging schemes gradually increases as the number of nodes increases because the number of charging-needed nodes increases, which leads to an increase in the number of nodes that need charging from the MC. Therefore, the time at which the charging-needed node receives the charging response increases accordingly. However, the charging latency of the RBF-EHAOCS is significantly lower than the other two schemes, because the scheme prefers to select the charging-needed node with the shortest charging time as the next charging node in the charging candidate set *Z*, which markedly reduces the waiting time of other charging-needed nodes.

The change in the number of nodes causes the overall performance of the network to change. As shown in Figure 7c, as the number of network nodes increases to below 75, the energy hole rate of nodes under the RBF-EHAOCS method and the charging latency of the charging-needed node are both in a state of slow growth. When the number of nodes is above 100, the charging load of the MC continuously increases due to the increase in the charging-needed nodes in the network, and the energy hole rate of nodes and the charging latency of the charging-needed nodes considerably increase. Therefore, when the number of nodes is 75–100, the number of nodes with energy holes and the charging latency of the charging-needed nodes are smaller, and the charging response fairness is relatively better.

### 5.4. Network Performance under Different MC Energy Capacities

The other parameters were kept constant, and the MC energy capacity was gradually increased from 100 to 8000 J to explore the influence of MC energy capacity on the performance of the scheme. Figure 8a–c show the network performance changes.

When the MC energy capacity is 100 J, since the MC energy is too low, few nodes can receive charging service in one energy cycle, and the nodes in the service pool *S* that require frequent charging show energy holes before the charging service are obtained (Figure 8a). Among them, the node energy hole ratio under the RBF-EHAOCS is the lowest, approximately 72.6%. As the energy that the MC can carry increases, the energy hole rate of the node drops sharply. Given the increase in MC energy capacity, the MC can provide a charging service for more nodes in one energy cycle. The node can receive the charging service within the interval of the maximum charging waiting time, and the number of energy holes in the node suddenly drops. When the MC carrying energy is greater than 1000 J, the energy hole rate of nodes continues to decrease, but the trend gradually slowed, and the MC energy capacity gradually stabilizes after reaching 3000 J. Some nodes remain far away from the MC because the MC energy capacity is large enough, although the MC has enough energy to provide charging services for the nodes. The charging response cannot be obtained in time with the charging planning due to the influence of the number of nodes or the charging efficiency.

The charging latency variation trend of the charging-needed node is similar to that of the energy hole rate of nodes, as shown in Figure 8b. When the MC energy capacity is 1000 J or less, it sharply rises, and the MC energy capacity tends to slowly increase at 1000 J or more. This mainly occurs, because the increase in the MC energy capacity means thatthe MC needs to plan the charging of more nodes in one energy cycle. The node that cannot be provided with the charging service due to insufficient MC energy can now receive the charging service, and the charging waiting time of the node will be longer, which results in an increasing charging latency for the entire node to be charged. In the whole process of MC energy capacity increase, the energy hole rate of nodes of RBF-EHAOCS and the charging latency of the charging-needed node are better than NJNP and SAMER, because the MC always appears as a node when the MC energy is sufficient. The minimum energy hole principle is used to select the next charging node, and the requested charging threshold is estimated while using the dynamic energy consumption rate predicted by the RBF. The charging request of the charging-needed node is also met within the interval of the maximum charging waiting time. The scheme fully considers the minimum time required for the MC to complete charging after the next charging node is selected to minimize the energy hole rate, so that the charging latency phase of the node in the charging service pool *S* is satisfied and is shorter than in the other two schemes.

Figure 8c shows that as the MC energy capacity increases, the energy hole rate of nodes under the RBF-EHAOCS scheme significantly decreases, and the charging latency of the charging-needed node markedly increases. After the MC energy capacity reaches 1000 J, the trend changes: it slows down and eventually stabilizes. Therefore, when the MC energy capacity is between 5000 and 8000 J, the probability of energy hole occurrences is the smallest under the proposed scheme, and the charging latency is basically stable, the network charging planning is more reasonable, and the reliability of the charging scheme is higher.

### 5.5. Network Performance under Different Charging Rates

The other parameters were maintained at the default value and the charging efficiency was gradually increased from 100 to 300 mJ/s to explore the effect of charging efficiency between MC and node on the performance of the scheme. Figure 9a–c show the network performance changes.

Figure 9a shows that the node energy hole ratios of the three charging schemes are relatively large when the charging efficiency is 100 mJ/s. However, the energy hole rate for the RBF-EHAOCS scheme is the lowest at 39.3%. As the charging efficiency gradually increases, the energy hole rate decreases. When the charging efficiency increases, the time that is required for the MC to charge the nodes decreases, as does the number of nodes that the MC can serve in one cycle of charging the node. Increases cause the charging node to decrease the time interval from the sending of the charging request to the completion of charging by the MC, and more quickly responds to the charging request of the charging-needed node, thereby continuously reducing the node energy hole rate.

In this process, we compared our RBF-EHAOCS scheme with the other two schemes. The energy hole rate of nodes is lower because the dynamics of the node energy consumption is fully considered, and the number of nodes in the service pool *S* decreases, which minimizes the system energy hole rate, which is always selected as the next charging node. When the charging efficiency is 275 mJ/s, the node energy hole rate for RBF-EHAOCS infinitely approaches zero. At this time, the nodes in the charging service pool in the network can obtain the charging service provided by the MC before an energy hole occurs in the node. The survival rate of the network node approaches one.

As shown in Figure 9b, the charging latency of the charging-needed node under the three schemes decreases with an increasing charging efficiency, and the time that is required to select the charging in the charging candidate set *Z* is the shortest when the MC energy is sufficient. The scheme of selecting the node as the next charging node ensures that the RBF-EHAOCS has smaller charging latency.

Figure 9c depicts the overall performance of the network under different charging efficiencies. With increasing charging efficiency, the energy hole rate of nodes under the RBF-EHAOCS scheme considerably decrease, while the charging latency of the charging-needed node slightly decreases when charging. After the efficiency reaches 175 mJ/s, the energy hole rate of nodes slowly decreases, and the charging latency of the charging-needed node decreases. The energy charging rate at the charging efficiency is 275 mJ/s, and the charging latency of the charging-needed node reaches the maximum. Therefore, when the charging efficiency is between 175 and 250 mJ/s, the energy hole probability of the node is low under this scheme, and the time that is required for the charging node to receive the charging service within the maximum charging waiting time is the least, and the network performance is relatively high.

## 6. Conclusions

In this study, we addressed the online dynamic charging problem of wireless rechargeable sensor networks and proposed an online charging scheme RBF-EHAOCS that was based on the RBF neural network for energy starvation avoidance. The scheme can predict the dynamics and diversity of sensor node energy consumption in an actual environment while using the RBF neural network, and fully considers the real-time selection of a charging request threshold and improves the fairness of the charging response and system stability. The next charging node is reasonably selected by carrying a single MC with limited energy, and energy holes in the charging-needed node are avoided as much as possible. In the simulation experiment, the performance of the RBF-EHAOCS was evaluated by analyzing the energy hole rate of nodes and the charging latency of the nodes needing charging. The experimental results showed that the charging scheme can effectively reduce the energy hole rate and quickly and effectively replenish the charging-needed nodes in the network. However, since the scheme mainly aims to reduce the energy hole rate, if a farther node is selected as the next charging node may occur, the MC charging cost might increase (the MC charging cost is defined as the total distance that the MC moves during charging). In subsequent research, dual MC co-charging could be considered, and a new auxiliary MC could assist in the response to the charging request of nodes, which could effectively reduce the MC charging cost while also reducing the node energy hole rate.

## Figures and Tables

**Figure 1 sensors-20-00205-f001:**
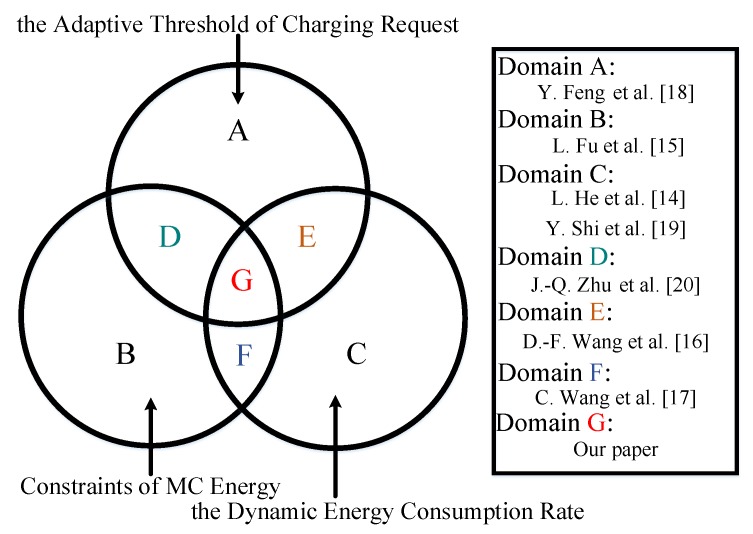
Summary of related works.

**Figure 2 sensors-20-00205-f002:**
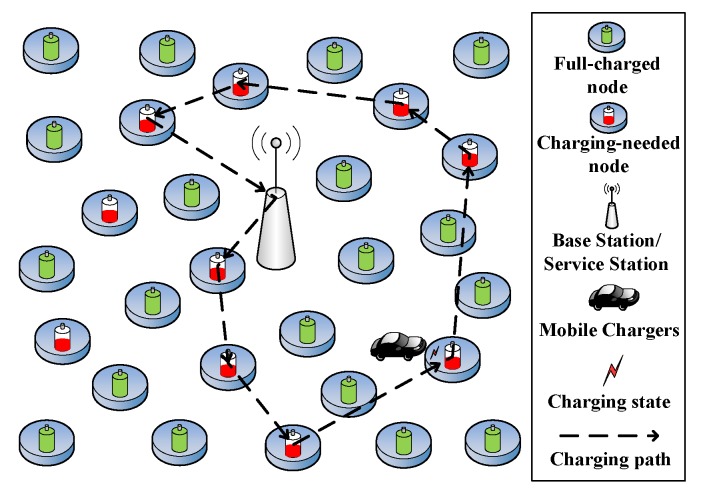
Network architecture.

**Figure 3 sensors-20-00205-f003:**
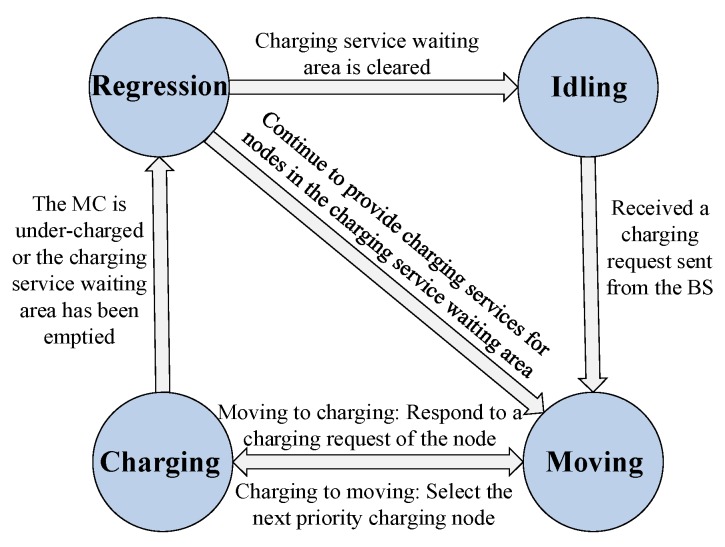
State transmission of the mobile charger (MC).

**Figure 4 sensors-20-00205-f004:**
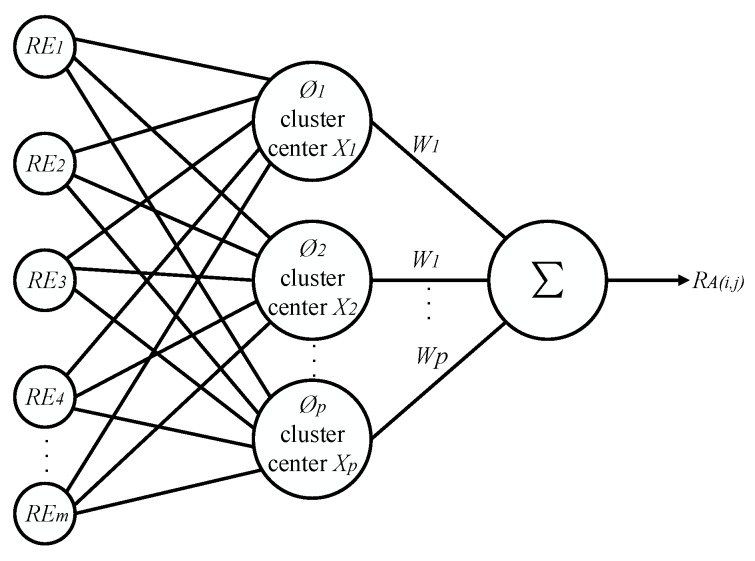
The network topology of the radial basis function (RBF) neural network.

**Figure 5 sensors-20-00205-f005:**
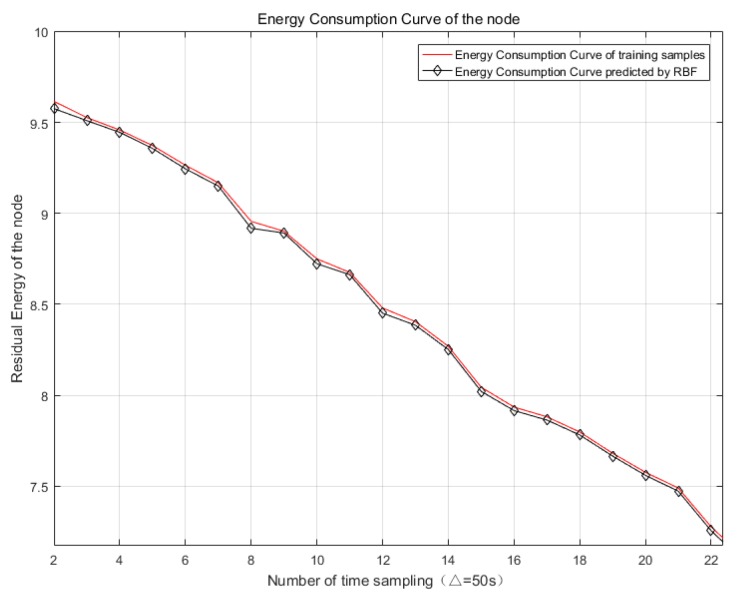
The dynamic energy consumption rate curve predicted by RBF.

**Figure 6 sensors-20-00205-f006:**
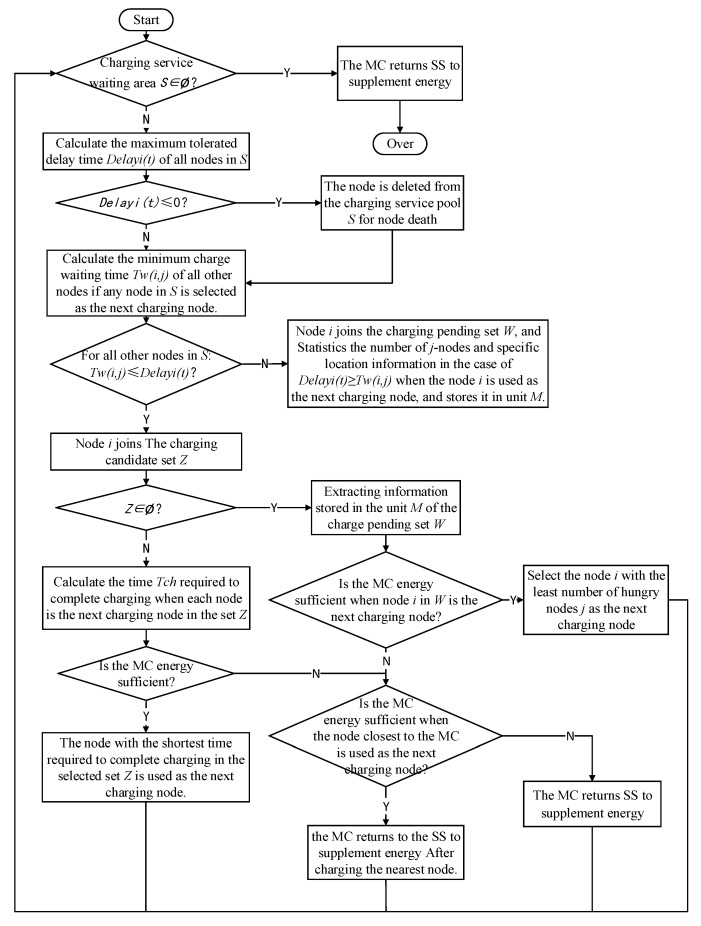
The flow chart of the next charging node selection scheme.

**Figure 7 sensors-20-00205-f007:**
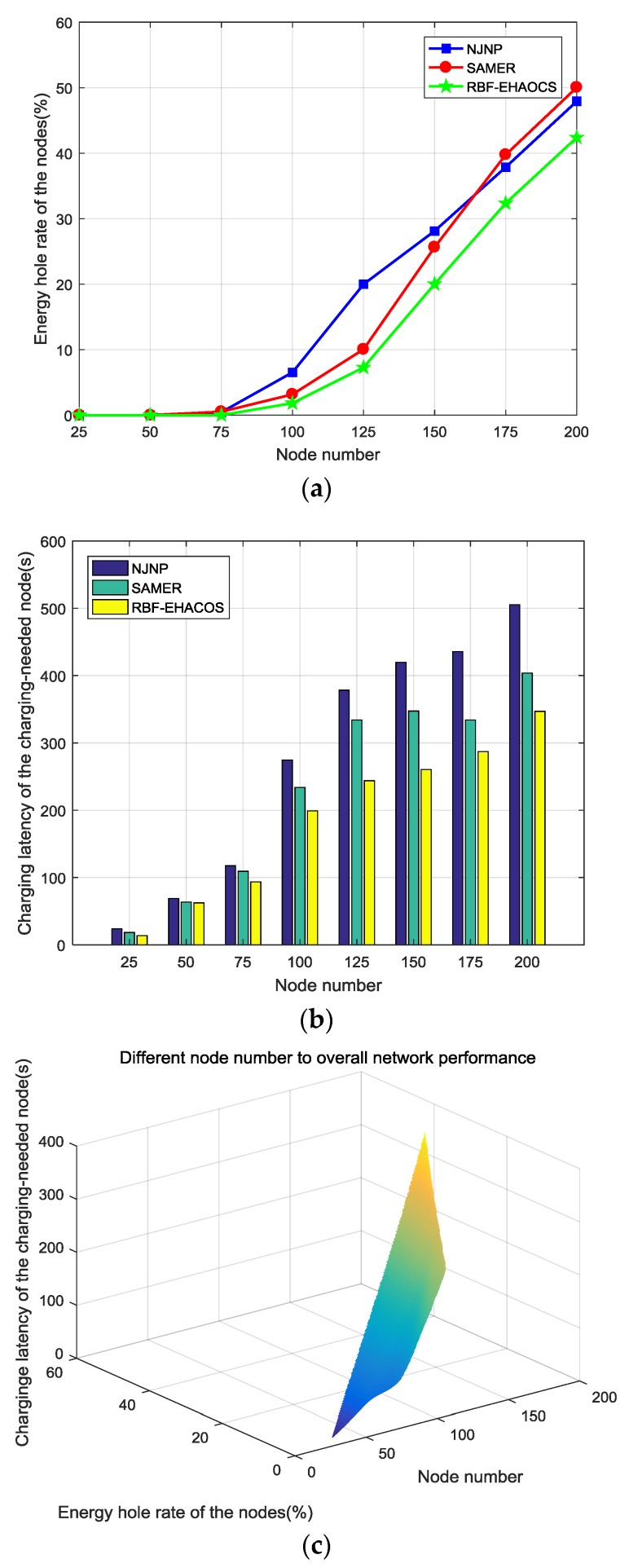
(**a**) Energy hole rate of the nodes, (**b**) charging latency of the charging-needed nodes, and (**c**) overall network performance.

**Figure 8 sensors-20-00205-f008:**
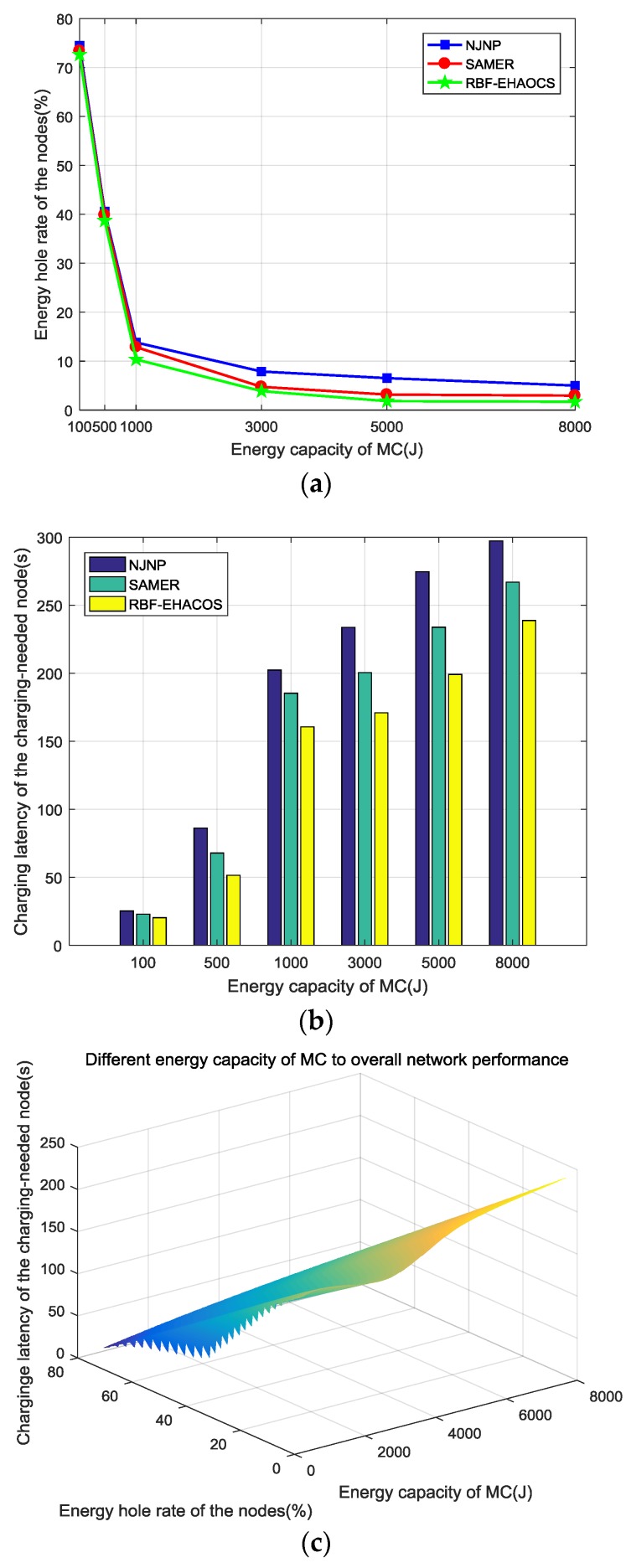
(**a**) Energy hole rate of the nodes, (**b**) charging latency of the charging-needed node, and (**c**) overall network performance.

**Figure 9 sensors-20-00205-f009:**
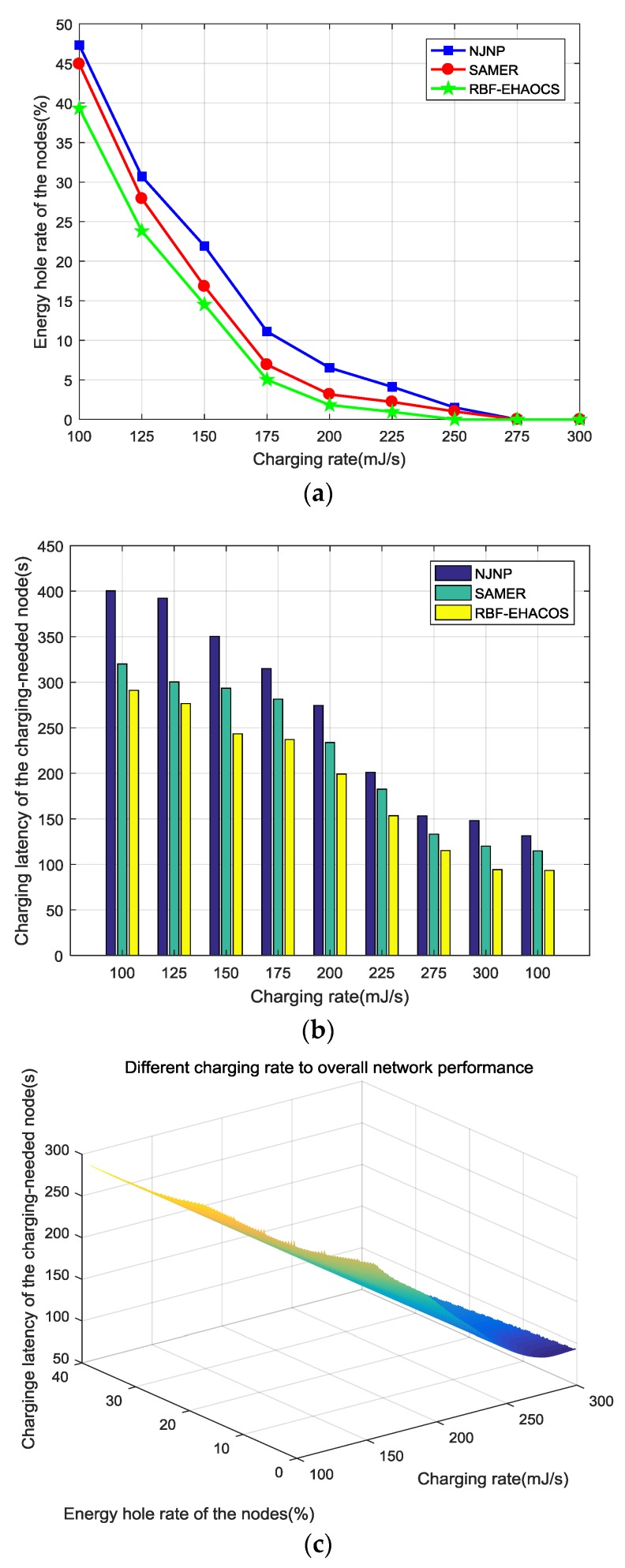
(**a**) Energy hole rate of the nodes, (**b**) charging latency of the charging-needed node, and (**c**) overall network performance.

**Table 1 sensors-20-00205-t001:** Experiment parameter settings.

Parameter	Value
Energy capacity of sensor nodes E	10 J
Energy capacity of MCs P	5000 J
Moving velocity of MCs v	5 m/s
Charging efficiency of MCs η	200 mJ/s
Energy consumption of MCs for moving a unit distance c	4 mJ
Expected energy use of MCs α	0.8
Energy consumption for receiving a packet at sensors $E_{recieve}$	0.4 mJ
Energy consumption for delivering a packet at sensors $E_{send}$	0.5 mJ
Propagation radius for sensor nodes R	25 m
Interval of each transmission Δ	10 s

**Table 2 sensors-20-00205-t002:** The performance under different schemes.

Scheme	Energy Hole Rate of Nodes (%)	Charging Latency of Charging-Needed Node (s)
NJNP	6.53	274.55
SAMER	3.17	233.86
RBF-EHAOCS	1.83	199.13

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
