# Peer review of "An Online Charging Scheme for Wireless Rechargeable Sensor Networks Based on a Radical Basis Function"

_sensors, 2019, doi:10.3390/s20010205_

Round 1
Reviewer 1 Report
The scheme uses the RBF neural network to predict the dynamic energy consumption rate during the charging process, and then estimates the optimal
threshold value of the node charging request on this basis, and then determines the next charging node by the selective conditions: the minimum energy hole rate and the shortest charging completion time.
The RBF is used to predict the dynamic energy consumption rate, so as to improve the real-time and fairness of the node charging response but it not clear that why would RBF be chosen as a prediction strategy? Why not use evolutionary neural networks or simple weighted predictors?
The next charging node is selected based on the minimum energy hole rate and the shortest charging completion time. Shorted charging time can vary a lot due to physical characteristics of the environment and hardware under use. Has any experiment considered this variance?
You need to discuss some real life examples for the implementation of MC's and the experimentation setup. There should be some rational behind the experiment setup.
Author Response
The scheme uses the RBF neural network to predict the dynamic energy consumption rate during the charging process, and then estimates the optimal threshold value of the node charging request on this basis, and then determines the next charging node by the selective conditions: the minimum energy hole rate and the shortest charging completion time. For the theoretical basis of selecting the radial basis function (RBF) neural network as the prediction strategy, we supplemented the explanation in the paper. For the prediction of dynamic energy consumption rate, the available methods include weighted average method, RBF neural network and the evolutionary neural network. The weighted average method is more focused on long-term trend changes, which is not conducive to the timeliness of calculation. In an actual WSN, the energy consumption of some nodes acting as the main cluster head changes more, which may lead to a large deviation between the predicted value and the actual value when using this method. The evolutionary neural network is an organic fusion of evolutionary computing and neural networks, which is more suitable for intelligent learning and prediction in the context of large systems and big data. Complex and time-consuming problems with adaptive processes are prone to occurring in the WSN with small data volumes and small network scales, which may prevent the real-time performance of the algorithm from being guaranteed. The RBF neural network is used as a nonlinear multi-layer single feedforward local approximation network that has strong robustness, nonlinear fitting ability, and memory ability. Compared with other two predictive learning methods, RBF has simpler learning rules, shorter prediction time, and better fitting effect on energy consumption. Therefore, we chose the RBF neural network to predict the dynamic energy consumption rate of the nodes. In subsequent research, we will consider improving the RBF neural network, such as using the PSO-RBF method to predict the dynamic energy consumption rate, and verify its feasibility.
When the physical characteristics of the experimental environment and the hardware used in the experiment are the same, it can be seen from the simulation results in this paper that, compared with the more traditional and popular NJNP and SAMER strategies, the energy hole rate of nodes and the charging delay of the charging-needed nodes of RBF-EHAOCS, which is proposed by us, are greatly reduced, and the optimization effect of the online charging strategy is more obvious. For the impact of the physical characteristics of the environment and hardware in the actual WRSN network, we did not consider it in the algorithm, but there are three factors that may have a greater impact on the algorithm performance of the online charging strategy in the actual network that, the number of nodes, the energy capacity of the MC and the charging rate between the MC and the node. Therefore, we perform a comparative analysis of them, in order that the optimization effect of RBF-EHAOCS in extending the network life cycle and reducing the charging waiting time is verified in various aspects.
Among the existing online charging strategies for wireless rechargeable sensor networks, most of them focus on longitudinal research in theory. The focus of our paper is also on theoretical and simulation research of the online charging strategy. The simulation parameters are set after reading a lot of literature, considering the compatibility of the parameters of each algorithm and the comparability of strategies’ performance. In order to avoid irrational parameter setting from affecting the optimization effect of the algorithm, we also performed simulation comparison on three important parameters, and confirmed the environmental parameters with the highest adaptability of our proposed algorithm. At the same time, our research team has started to build the corresponding WRSN hardware platform, and has completed the Ad hoc network of network nodes, and they can successfully communicate with the host computer now. In the subsequent research work, we will continue building the hardware platform, and working for the actual experimental verification of the RBF-RHAOCS strategy and the impact of different hardware designs.
Submission Date: 20 December 2019
Reviewer 2 Report
*Introduction is vaguely written. My suggestion is to divide the introduction into three subsections: 1) motivation and incitement, 2) literature review and 3) contribution and paper organization.
*The authors should explain what are the novel aspects of their paper in relation to the current literature. It lacks a clear comparison between the submitted paper and the more relevant literature contributions, which should highlight the main advantages of the current submission.
*Improve quality of the Figures.
*Simulation Results shall be enriched.
*I would advise that the authors get a native English speaker to review the paper for grammatical issues.
Author Response
We reorganized the presentation and related work and re-edited it, and now we divide the introduction into three subsections: 1) motivation and incitement, 2) literature review and 3) contribution and paper organization. In terms of introduction, we derived the offline charging strategy and online charging strategy from the rise of wireless rechargeable sensor networks firstly. Then, we analyzed the problems existing in the online charging strategy and proposed the RBF-EHAOCS strategy and explained its four main contributions. In terms of related work, the three indicators that classify the algorithms mentioned in the references are used as the main line, the adaptive threshold of charging request, the constraints of MC energy, and the dynamic energy consumption rate. And the common algorithms in online charging strategies are described, and the innovation of this article is highlighted. Finally, the three questions solved in this question are explained, and the paper organization is supplemented.
In terms of related work, we classify the algorithms mentioned in the references with three indicators, the adaptive threshold of charging request, the constraints of MC energy, and the dynamic energy consumption rate, and use them to run through the entire line of cited references and innovations in this article. At the same time, in the introduction of the article, the three main problems solved and the four main contributions made in this article are listed. The algorithms proposed in this paper all meet the requirements of the three indicators to achieve the goal of the minimum energy hole rate of the node and the shortest charging waiting time of the charging-needed node.
For the quality of the Figures, we re-copy and paste directly from the simulation software through simulation graphics editing, and the Figures have higher definition now. For the simulation result of the charging delay of the charging-needed node, the histogram is used to compare and analyze the simulation results of each algorithm more intuitively.
For the simulation results, we added a comparison simulation of the energy hole rate of nodes of each algorithm and the charging delay of the charging-needed node under the preset initial simulation conditions, and explained it. Meanwhile, we conducted a comparative analysis of the three influencing factors, that may affect the performance of the online charging strategy in an actual WRSN network, the number of nodes, the capacity of the MC to carry the energy, and the charging efficiency of the MC. The optimization effect of RBF-EHAOCS in extending the network life cycle and reducing the charging waiting time is shown more comprehensively. For grammatical issues, we have applied for the English editing service of MDPI, the service number is English-14945. The revision has been made to the comments in the edited English manuscript.
Submission Date: 20 December 2019
Round 2
Reviewer 2 Report
I am satisfied with the corrections that introduced by the authors